# Constraint of Base Pairing on HDV Genome Evolution

**DOI:** 10.3390/v13122350

**Published:** 2021-11-23

**Authors:** Saki Nagata, Ryoji Kiyohara, Hiroyuki Toh

**Affiliations:** Department of Biomedical Chemistry, School of Science and Technology, Kwansei Gakuin University, Gakuen, Sanda 669-1337, Japan; gsu12514@kwansei.ac.jp (S.N.); dcc13727@kwansei.ac.jp (R.K.)

**Keywords:** hepatitis delta virus, hepatitis delta antigen, base pairing, one-sample Wilcoxon test

## Abstract

The hepatitis delta virus is a single-stranded circular RNA virus, which is characterized by high self-complementarity. About 70% of the genome sequences can form base-pairs with internal nucleotides. There are many studies on the evolution of the hepatitis delta virus. However, the secondary structure has not been taken into account in these studies. In this study, we developed a method to examine the effect of base pairing as a constraint on the nucleotide substitutions during the evolution of the hepatitis delta virus. The method revealed that the base pairing can reduce the evolutionary rate in the non-coding region of the virus. In addition, it is suggested that the non-coding nucleotides without base pairing may be under some constraint, and that the intensity of the constraint is weaker than that by the base pairing but stronger than that on the synonymous site.

## 1. Introduction

Hepatitis D is a form of human liver disease caused by the hepatitis delta virus (HDV) [1]. HDV has been identified as a satellite virus of the hepatitis B virus (HBV). The virion assembly and infectivity of HDV depend on HBV [2,3]. HDV was originally classified into three genotypes: I, II, and III [4,5,6]. Genotype II is further divided into two subtypes, IIa and IIb. The subsequent accumulation of sequence data and the molecular phylogenetic analysis has revealed that HDV consists of eight clades, HDV-1~-8 [6,7]. HDV-1~-4 correspond to Genotypes I, IIa, III, and IIb. In the recent analysis of HDV full-length genomes, the eight clades were reorganized into three groups [8]. HDV has been mainly identified from human patients. Recently, however, hepatitis D–like agents have been identified from birds, newts, toads, fish, and termites [9,10].

HDV is a single-stranded circular RNA virus. The genome size of HDV is about 1700 bases, and about 70% of the genome is self-complementary. The genome is folded into an unbranched rod-like structure by the base pairings of the complementary region [11,12]. The genome has a viroid-like domain [13,14]. Each of the genomic and the antigenomic sequences of the viroid-like domain include a cleavage site by the ribozyme mechanism [15,16,17]. A protein-coding gene is also encoded in the genome, from which two isoforms of proteins are yielded. One of them is called small delta antigen (S-HDAg), which is a protein of about 195 aa in length. The amber stop codon of the S-HDAg gene is changed to encode tryptophan by the RNA editing [12,18], which extends the C-terminal region by 19 aa. The longer isoform is called long delta antigen (L-HDAg). The S-HDAg is required for the genomic replication of HDV [19], whereas the L-HDAg represses the replication [20,21] and is involved in the packaging of the viral particle [22,23].

The molecular evolution of HDAg has been investigated with the amino acid and nucleotide sequences. Several groups have estimated the evolutionary rate of HDV [24,25,26]. The sequences obtained from the same patients at different times were used in the studies. The three groups did not distinguish the synonymous substitutions from the non-synonymous substitutions in their analyses. The orders of most of the estimated evolutionary rates are 10^−3^ per nucleotide site per year. Chao, Tang, and Hsu (1994) found that the evolutionary rates of different parts of the HDV genome vary [26]. Krushkal and Li (1995) estimated the evolutionary rates of the coding and non-coding region of HDV [27]. In their study, the synonymous and non-synonymous evolutionary rates were estimated for the coding region. The orders of the estimated evolutionary rates are the same as those of previous studies. The synonymous evolutionary rates are smaller than the non-synonymous evolutionary rates and the evolutionary rates of the non-coding region. They considered that the strong GC bias at the third codon position is the cause of the low rates of the synonymous substitutions. Wu et al. (1999) identified nine segments of the HDV genome with high levels of recombination [28]. They suggested that the recombination can contribute to the genetic diversity of HDV. Anisimova and Yang (2004) examined 33 sequences of the HDAg gene from Genotypes I, II, and III to study the diversifying selection on the HDAg gene [29]. Then, it was found that about 11% codon sites of the HDAg gene are under positive selection. No significant evidence for recombination was obtained from their study. Bishal, Mukherjee, and Chakraborty (2013) investigated the synonymous codon usage pattern of the HDAg gene to find that the most important determinant of the codon usage is the mutation pressure [30]. Delfino et al. (2018) reported comprehensive computational analysis of HDV full-length genomes, in which phylogenetic relationships and various characteristics of the amino acid sequences are described [8].

Thus, there are many studies on the molecular evolution of HDV. In particular, HDAg has been often used as the subject of evolutionary studies. However, the secondary structure of the genome has not been explicitly taken into account in such studies. As described above, about 70% of the HDV genome is self-complementary. If the folding of the HDV genome into a secondary structure is important for the activity of HDV, the base pairings would function as the constraint to reduce the nucleotide substitution rate. In this study, we developed a new method based on a one-sample Wilcoxon test to examine the effect of the base pairing as the constraint on the nucleotide substitution. The result of our analysis suggests that the base pairing imposes constraint on the nucleotide substitution in the non-coding region of HDV. It is also suggested that the nucleotides in the non-coding region that are not involved in the base pairing may be under some constraint. The intensity of the constraint on the non-coding region without base pairing is weaker than that on the non-coding region with base pairing but stronger than that on the synonymous site.

## 2. Materials and Methods

### 2.1. Alignment of HDV Genomes and L-HDAg Genes

We collected 498 genome sequences from NCBI (https://www.ncbi.nlm.nih.gov, 2 September 2021). Because of the circular structures, the sequences thus obtained did not always start from an identical position. Therefore, a preliminary alignment was made with MAFFT [31], a multiple alignment tool, to identify the conserved sequences, which could be used as a landmark to adjust the positions to start the alignment. The consensus sequence of the landmark region thus obtained was GCCCAGGUCGGACC, where the invariant nucleotides in the consensus sequence were underlined. All the genome sequences were edited to start from the region corresponding to the consensus sequence. The sequences thus reconstructed were again aligned with MAFFT. The sequences with long gaps and unidentified nucleotides were excluded from the sequence set. The sequences including frameshifts in the coding region were also excluded. Then, 226 genome sequences were obtained. A final alignment of the 226 genome sequences was made with an option of MAFFT, mafft-qinsi --debug, in order to take the secondary structure information into account [32]. The option invokes secondary structure prediction by the McCaskill algorithm [33], implemented in the Vienna RNA package [34]. In the algorithm, the probabilities of the secondary structures are calculated based on the free energies at first. Out of the possible secondary structures, the probabilities of the structures, in which a pair of bases are bound to each other, are summed. The sum is the base-pairing probability of the two bases [33]. The base-pairing probabilities are used to modify the similarity score between bases of different sequences to improve the accuracy of the alignment [32]. The complete 3D structure of the HDV genome is not available yet. So, in this study, the probability was used to distinguish the bases that bind to other bases from the bases that are not involved in base pairing, as described below. The alignment file of the FASTA format is given in Appendix A.

The nucleotide sequences corresponding to the L-HDAg gene were extracted from the genome alignment. The complementary sequences of the extracted sequences were generated to obtain the ORFs for HDAg, because the HDV genome is a negative sense RNA. The gaps were excluded from the complementary sequences, and the unaligned sequences were translated into amino acid sequences. A multiple alignment of the amino acid sequences was generated with MAFFT. The option, --auto, was used for the alignment. The amino acid sequence alignment was converted into a nucleotide sequence alignment of the L-HDAg coding regions, with the tranalign function of EMBOSS [35]. The alignment file of the FASTA format is given as Appendix A. A phylogenetic tree of the aligned amino acid sequences was generated by the neighbor-joining method [36] with MEGA X version 10.2.4 [37]. The JTT + G model was used for the calculation of the evolutionary distance for every possible pair of the aligned amino acid sequences. All the sites including gaps were excluded from the alignment to calculate the distances. According to the tree, the aligned sequences were classified into three groups corresponding to Genotypes I, II, and III [11,38,39]. The GenBank IDs for the 226 genome sequences used for the comparative study are listed in Table 1.

### 2.2. Threshold for Base-Pairing Probability

As described above, MAFFT [32] uses the McCaskill algorithm [33] to incorporate the base-paring probability to improve the alignment accuracy. In this study, the base-pairing probabilities obtained by the alignment operation were used for the analysis. To predict the bases that are involved in the base pairing, the threshold value for the base-paring probability was examined. We calculated the following value for each nucleotide in the genome sequence.
*ps*(*i*, *j*) = max{base-pairing probability(*i*, *j*, *k*)|*i* and *j* are fixed}
where *j* and *k* indicate nucleotide positions in the *i*-th genome sequence. The argument of the max operation is the base-paring probability between bases *j* and *k* of the *i*-the genome sequence. The operation finds the maximum base-pairing probability under the condition that the indices, *i* and *j* are fixed. Then, the number of bases with the *ps* value greater than a given threshold over the total number of bases was calculated for each genome sequence. The ratio means the predicted fraction of the bases involved in binding to other bases in a genome sequence under the threshold value. The averages of the ratio for different threshold values over the 226 sequences are shown in Table 2.

As shown in the table, about 70% of the bases of the HDV genome are predicted to be involved in base pairing under the threshold value = 0.5. Therefore, 0.5 was used as the threshold value of base-pairing probability in this study.

The threshold 0.5 was also used to define the complementary region of the coding region. The region that is complementary to the coding region is considered to be subject to the constraint at protein level through the binding to the bases in the coding region. So, the complementary region should be excluded from the non-coding region. The alignment sites 110–793 in the Appendix A correspond to the coding region of L-HDAg. When more than 70% of the bases at an alignment site had the base-pairing probability to any base in the coding region, the site was regarded as a constituent of the complementary region. Then, the alignment sites 912–1932 were identified as the complementary region, although the region included the sites that did not satisfy the criterion of 70%. So, the three regions of the alignment sites, 1–109, 794–911, and 1933–2211 were used as the non-coding region. The positions of the coding region and the non-coding region are shown in the schematic diagram of the genome alignment (see Figure 1).

### 2.3. Calculation of Evolutionary Distance

The evolutionary distance between every pair of aligned sequences was calculated for the non-coding region defined above. Consider a pair of sequences *x* and *y*. Let *N*(*x*, *y*) be the number of sites without gaps in the non-coding region of sequences *x* and *y*. Let *n*(*x*, *y*) be the number of sites occupied by different bases between sequences *x* and *y* in the *N* sites. The sequence difference between *x* and *y* was calculated as *n*(*x*, *y*)/*N*(*x*, *y*). The evolutionary distance between *x* and *y*, *ed*(*x*, *y*), was obtained by correcting the effect of multiple hits with a nucleotide substitution model. There are many nucleotide substitution models for the correction. Out of them, the Jukes and Cantor nucleotide substitution model [40] was used in this study, because the model was adopted by SNAP [41] as described below. The following is the equation for the correction.
*ed*(*x*, *y*) = −(3/4)log(1 − (3/4)(*n*(*x*, *y*)/*N*(*x*, *y*)))

In addition to the distance for the entire non-coding region, the distances for base-pairing sites and those for non-base-pairing sites in the non-coding region were calculated. The base-pairing and non-base-pairing sites were defined for each pair of aligned sequences. Consider a pair of aligned sequences *x* and *y*, the *z*-th alignment site, and a function *b*(*x*, *z*), which returns the nucleotide position in the sequence *x* corresponding to the *z*-th alignment site. When both *ps*(*x*, *b*(*x*, *z*)) and *ps*(*y*, *b*(*y*, *z*)) were greater than 0.5, the *z*-th alignment site was regarded as a base-pairing site for the *x*-th and the *y*-th sequences. Inversely, the *z*-th alignment site was regarded as a non-base-pairing site for sequences *x* and *y*, if both *ps*(*x*, *b*(*x*, *z*)) and *ps*(*y*, *b*(*y*, *z*)) were less than or equal to 0.5. The evolutionary distances for the base-pairing and non-base-pairing sites in the non-coding region for sequences *x* and *y* were calculated by the same manner as that for the entire non-coding region, which are denoted as *ed*_bp_(*x*, *y*) and *ed*_nbp_(*x*, *y*).

Two types of the evolutionary distances, synonymous and non-synonymous distances, between sequences *x* and *y* were calculated for the coding region of L-HDAg. The former is a distance based on the nucleotide substitutions that do not change the encoded amino acid, whereas the latter is a distance based on the nucleotide substitutions to induce the amino acid substitutions [42]. At first, the number of the synonymous substitutions per site ($K_{S}\left(x,y\right)$) and the number of the non-synonymous substitutions per site ($K_{a}\left(x,y\right)$) were calculated by the Nei-Gojobori method [43]. The effect of the multiple hits was corrected by the Jukes and Cantor nucleotide substitution model [40]. The corrected $K_{S}\left(x,y\right)$ and $K_{a}\left(x,y\right)$ were used as the evolutionary distances of the coding region between sequences *x* and *y*, which are denoted as $K_{s}^{c}\left(x,y\right)$ and $K_{a}^{c}\left(x,y\right).\;$A perl program, SNAP [41], was used for the calculation.

### 2.4. Analysis of Distance Ratio

As described above, five distances were calculated for different regions of the alignment. The constraint of a region is considered to determine the molecular evolutionary rate of the region. A strong constraint would decrease the evolutionary rate, and vice versa. Consider two regions in HDV genome. One of the regions has a large evolutionary rate, whereas the evolutionary rate of the other is small. Then the evolutionary distance of the former would be larger than that of the latter when the two sequences are compared. Under this assumption, the constraint by the base pairing was investigated (see Figure 2).

Suppose that two regions in the alignment, *a* and *b*, have the molecular evolutionary rates *v_a_* and *v_b_*, and that a pair of aligned sequences, *x* and *y*, diverged from the common ancestral sequence at time *t_ab_*. For simplicity, let us assume that the evolutionary rates did not change during the time after the divergence from the ancestral sequence. Then, the evolutionary distances of regions *a* and b between *x* and *y*, *d_a_*(*x*, *y*) and *d_b_*(*x*, *y*), are expressed as follows (see Figure 2a):$$d_{a}\left(x,y\right)≒2t_{xy}\cdot v_{a}$$
$$d_{b}\left(x,y\right)≒2t_{xy}\cdot v_{b}$$

The nearly equal symbol is used for the formula instead of the equal symbol because the molecular evolution is a stochastic event. The ratio of the two distances is expressed as the rate ratio.
$$\frac{d_{a}\left(x,y\right)}{d_{b}\left(x,y\right)}\;≒\;\frac{v_{a}}{v_{b}}$$

The rate ratio is considered to reflect the difference in constraint between the two regions. If the constraint on the region *a* is stronger than that on the region *b*, the ratio would take a value less than 1.0. Inversely, the ratio would take a value greater than 1.0 if the constraint on the region *a* is weaker than that on region *b*. The point of the operation is that taking the ratio can cancel the divergence time. Consider the other pair of aligned sequences, *z* and *w*. The distances for the regions, *a* and *b*, between the sequences are expressed as follows:$$d_{a}\left(z,w\right)≒2t_{zw}\cdot v_{a}$$
$$d_{b}\left(z,w\right)≒2t_{zw}\cdot v_{b}$$

The rate ratio is obtained again by taking the ratio of the distances.
$$\frac{d_{a}\left(z,w\right)}{d_{b}\left(z,w\right)}\;≒\;\frac{v_{a}}{v_{b}}$$

So, we can sample the rate ratios by taking the distance ratios by changing the combination of sequence pairs (see Figure 2b). As described above, the evolutionary rate of a region is determined by the constraint on the region. So, if there is no difference in constraint between two regions, *a* and *b*, the evolutionary rates, $v_{a}$ and $v_{b}$, would be close to each other. Then, the median of the distance ratios calculated between various pairs of aligned sequences would take a value close to one.

Consider the base-pairing sites and non-base-pairing sites as the two regions in an alignment. Both sites belong to the non-coding region of the HDV genome. If base pairing does not function as the constraint on the base-pairing sites, there would be no difference in constraint between the base-pairing and non-base-pairing sites. As described above, the median of the distance ratios, *ed*_bp_(*x*, *y*)/*ed*_nbp_(*x*, *y*), calculated between every possible pair of aligned sequences except for the pairs with the distance *ed*_nbp_(*x*, *y*) = 0.0, is expected to take a value close to one. The one-sample Wilcoxon test can be applied to the data of the distance ratios, to examine whether the base-pairing functions as the constraint on the base-pairing sites (see Figure 2c). The null hypothesis of the test is that the median of the data = 1.0. If the base-pairing function is the constraint, the evolutionary rate of the base-pairing sites would be smaller than that of the non-base-pairing sites. So, the alternative hypothesis is set that the median of the data < 1.0.

In addition to *ed*_bp_(*x*, *y*)/*ed*_nbp_(*x*, *y*), nine other distance ratios, *ed*_bp_(*x*, *y*)/ed(*x*, *y*), *ed*_nbp_(*x*, *y*)/ed(*x*, *y*), *ed*_bp_(*x*, *y*)/$K_{s}^{c}\left(x,y\right)$, *ed*_nbp_(*x*, *y*)/$K_{s}^{c}\left(x,y\right),\;$*ed*_nbp_(*x*, *y*)/$K_{s}^{c}\left(x,y\right),$
*ed*_bp_(*x*, *y*)/$K_{a}^{c}\left(x,y\right)$, *ed*_nbp_(*x*, *y*)/$K_{a}^{c}\left(x,y\right),\;$*ed*_nbp_(*x*, *y*)/$K_{a}^{c}\left(x,y\right),\;$and $K_{a}^{c}\left(x,y\right)$/$K_{s}^{c}\left(x,y\right)$, were examined by the one-sample Wilcoxon test. The setting of the alternative hypothesis for each distance ratio is described in the legends for Table 3 and Table 4. The computational language for statistical analysis, R [44], was used for the one-sample Wilcoxon test. The effect of the multiple comparison on the *p*-values was corrected by the Bonferroni method. That is, the corrected *p*-values were obtained by multiplying the *p*-values by ten. The corrected *p*-values were examined with the significance level = 0.01. The distances within each of three Genotypes and those between every possible pair of the Genotypes were calculated, and the ten distance ratios were analyzed by the procedure described above.

## 3. Results

### 3.1. Comparison within Each Genotype

The evolutionary distances, *ed*_bp_(*x*, *y*), *ed*_nbp_(*x*, *y*), *ed*(*x*, *y*), $K_{s}^{c}\left(x,y\right),$ and $K_{a}^{c}\left(x,y\right),\;$between every possible pair of aligned sequences, except for the pair which has *ed*_nbp_(*x*, *y*) = 0.0, *ed*(*x*, *y*) = 0.0, $K_{s}^{c}\left(x,y\right)$ = 0.0, or $K_{s}^{c}\left(x,y\right)$ = 0.0, are shown in Appendix A, and the summary of the distances is shown in Appendix A. The pairs with *ed*_nbp_(*x*, *y*), *ed*(*x*, *y*), $K_{s}^{c}\left(x,y\right)$, or $K_{a}^{c}\left(x,y\right)$ = 0.0 were excluded from the analysis, because the four distances were used as the denominators of the distance ratios. The numbers of base-pairing sites, non-base-pairing sites, the entire non-coding region, synonymous sites, and non-synonymous sites are also shown in Appendix A. The box plots of the ten distance ratios calculated by the sequence comparison within each genotype are shown in Figure 3. As shown in the figure, the distance ratios of either genotype did not seem to follow the normal distribution. So, the distance ratios were examined by the one-sample Wilcoxon test (see Table 3).

The medians and corrected *p*-values for ten distance ratios, *ed*_bp_(*x*, *y*)/*ed*_nbp_(*x*, *y*), *ed*_bp_(*x*, *y*)/*ed*(*x*, *y*), *ed*_nbp_(*x*, *y*)/*ed*(*x*, *y*), *ed*_bp_(*x*, *y*)/$K_{s}^{c}\left(x,y\right)$, *ed*_nbp_(*x*, *y*)/$K_{s}^{c}\left(x,y\right),\;$*ed*_nbp_(*x*, *y*)/$K_{s}^{c}\left(x,y\right),$ *ed*_bp_(*x,y*)/$K_{a}^{c}\left(x,y\right)$, *ed*_nbp_(*x,y*)/$K_{a}^{c}\left(x,y\right),\;$*ed*_nbp_(*x*, *y*)/$K_{a}^{c}\left(x,y\right),$ and $K_{a}^{c}\left(x,y\right)$/$K_{s}^{c}\left(x,y\right),\;$calculated by the sequence comparison within each genotype are shown in Table 3. All the corrected *p*-values for the ten distance ratios were less than 0.01 for either Genotype I or II. In contrast, all the corrected *p*-values of Genotype III were greater than 0.01, which may be due to the small sample size (see Table 3). Therefore, only the distance ratios within Genotype I and those within Genotype II were examined. Out of the ten distance ratios, we focused on the six ratios, *ed*_bp_(*x*, *y*)/*ed*_nbp_(*x*, *y*), *ed*(*x*, *y*)/$K_{s}^{c}\left(x,y\right),$ *ed*_bp_(*x*, *y*)/$K_{s}^{c}\left(x,y\right)$, *ed*_nbp_(*x*, *y*)/$K_{s}^{c}\left(x,y\right)$, *ed*_bp_(*x*, *y*)/$K_{a}^{c}\left(x,y\right)$, and *ed*_nbp_(*x*, *y*)/$K_{a}^{c}\left(x,y\right)$, to examine the effect of the base pairing.

The constraint on the base-pairing sites was compared with that on the non-base-pairing sites. The null hypothesis that the median of *ed*_bp_(*x*, *y*)/*ed*_nbp_(*x*, *y*) is equal to 1.0 was rejected for either Genotype I or II. The alternative hypothesis was that the median of *ed*_bp_(*x*, *y*)/*ed*_nbp_(*x*, *y*) is less than 1.0. The results suggest that the constraint on the base-pairing sites is stronger than that on non-base-pairing sites. That is, base-pairing functions as the constraint on the non-coding region.

The *ed*(*x*, *y*) is the evolutionary distance of the entire non-coding region. The entire non-coding region consists of the base-pairing and non-base-pairing sites. The synonymous sites are also the mixture of the base-pairing and non-base-pairing sites. The corrected *p*-value calculated for the null hypothesis that the median is equal to 1.0 was less than 0.01 for either Genotype I or II. The alternative hypothesis was that the median of the distance ratio is less than 1.0. The results suggest that the constraint on the non-coding region is stronger than that on the synonymous sites. Furthermore, to investigate the contribution of base pairing to the constraint, the non-coding region was divided into base-pairing and non-base-pairing sites, and the distance ratios, *ed*_bp_(*x*, *y*)/$K_{s}^{c}\left(x,y\right)$ and *ed*_nbp_(*x*, *y*)/$K_{s}^{c}\left(x,y\right)$, were examined by the one-sample Wilcoxon test. The null hypothesis that the median of *ed*_bp_(*x*, *y*)/$K_{s}^{c}\left(x,y\right)$ is equal to 1.0 was rejected for either Genotype I or II. The alternative hypothesis was that *ed*_bp_(*x*, *y*)/$K_{s}^{c}\left(x,y\right)$ is less than 1.0. That is, the constraint on the base-pairing sites is stronger than that on the synonymous sites. The null hypothesis that the median of *ed*_nbp_(*x*, *y*)/$K_{s}^{c}\left(x,y\right)$ is equal to 1.0 was also rejected. The alternative hypothesis was that the median of *ed*_nbp_(*x*, *y*)/$K_{s}^{c}\left(x,y\right)$ is less than 1.0. That is, the constraint on the non-base-pairing sites is stronger than that on the synonymous sites.

The constraint on the base-pairing sites and that on the non-base-pairing sites were compared with the constraint on the non-synonymous sites with the two distance ratios, *ed*_bp_(*x*, *y*)/$K_{a}^{c}\left(x,y\right)$ and *ed*_nbp_(*x*, *y*)/$K_{a}^{c}\left(x,y\right)$. The null hypothesis that the median of *ed*_bp_(*x*, *y*)/$K_{a}^{c}\left(x,y\right)$ is equal to 1.0 was rejected for either Genotype I or II. The alternative hypothesis was that *ed*_bp_(*x*, *y*)/$K_{a}^{c}\left(x,y\right)$ is less than 1.0. This means that the constraint on the non-coding region is stronger than that on the non-synonymous sites. The null hypothesis that the median of *ed*_nbp_(*x*, *y*)/$K_{a}^{c}\left(x,y\right)$ is equal to 1.0 was also rejected. The alternative hypothesis was that *ed*_nbp_(*x*, *y*)/$K_{a}^{c}\left(x,y\right)$ is greater than 1.0. The results suggest that the constraint on the non-base-pairing sites is weaker than that on the non-synonymous sites.

### 3.2. Comparison between Genotypes

The box plots of the ten distance ratios calculated by the sequence comparison between every pair of three genotypes. Genotypes are shown in Figure 4. Like the distribution of the distance ratios within each genotype, the distributions of the ratios between genotypes did not seem to follow the normal distribution. So, the distance ratios were examined with the one-sample Wilcoxon test. The result is summarized in Table 4. As shown in the table, all the corrected *p*-values, except for that of *ed*_nbp_(*x*, *y*)/*ed*(*x*, *y*), between Genotypes I and II were less than $2.2\times 10^{-15}$. The corrected *p*-value corresponding to *ed*_nbp_(*x*, *y*)/*ed*(*x*, *y*) between Genotypes I and II was 1.8$\times 10^{-7}$. So, all the corrected *p*-values were less than 0.01. The results of the statistical analyses for the distance ratios between Genotypes were basically consistent with those within Genotype I or II. Out of the ten distance ratios, we here describe the results of the statistical analyses, focusing on the six ratios, *ed*_bp_(*x*, *y*)/*ed*_nbp_(*x*, *y*), *ed*(*x*, *y*)/$K_{s}^{c}\left(x,y\right),$ *ed*_bp_(*x*, *y*)/$K_{s}^{c}\left(x,y\right)$, *ed*_nbp_(*x*, *y*)/$K_{s}^{c}\left(x,y\right)$, *ed*_bp_(*x*, *y*)/$K_{a}^{c}\left(x,y\right)$, and *ed*_nbp_(*x*, *y*)/$K_{a}^{c}\left(x,y\right)$, to examine the effect of the base pairing.

The intensity of the constraint on the base-pairing sites relative to that on the non-base-pairing sites was examined. In the comparison between genotypes, the null hypothesis that the median of *ed*_bp_(*x*, *y*)/*ed*_nbp_(*x*, *y*) = 1.0 was rejected. The alternative hypothesis was that the median of *ed*_bp_(*x*, *y*)/*ed*_nbp_(*x*, *y*) < 1.0. The results suggest that the constraint on the base-pairing sites is stronger than that on non-base-pairing sites.

The constraint on the non-coding region was compared with that on the synonymous sites. The null hypothesis that the median of *ed*(*x*, *y*)/$K_{s}^{c}\left(x,y\right)$ = 1.0 was rejected. The alternative hypothesis was that the median of the ratios was less than 1.0. The results suggest that the constraint on the non-coding region is stronger than that on the synonymous sites. The non-coding region was divided into base-pairing and non-base-pairing sites, and the two different distance ratios, *ed*_bp_(*x*, *y*)/$K_{s}^{c}\left(x,y\right)$ and *ed*_nbp_(*x*, *y*)/$K_{s}^{c}\left(x,y\right)$, were examined. The corrected *p*-values for the null hypothesis that the median of *ed*_bp_(*x*, *y*)/$K_{s}^{c}\left(x,y\right)=1.0$ was less than 0.01 for either pair of the genotypes. The alternative hypothesis was that the median of the distance ratios is less than 1.0. This means that the constraint on the base-pairing sites is greater than that on the synonymous sites. The corrected *p*-values for the null hypothesis that the median of *ed*_nbp_(*x*, *y*)/$K_{s}^{c}\left(x,y\right)$ is equal to 1.0 were less than 0.01, for either pair of the genotypes. The alternative hypothesis was that the median of *ed*_nbp_(*x*, *y*)/$K_{s}^{c}\left(x,y\right)$ is less than 1.0. So, not only the constraint on base-pairing sites, but also that on the non-base-pairing sites, is stronger than that on the synonymous sites.

The constraints on the base-pairing sites and non-base-pairing sites were compared with that on the non-synonymous sites. The corrected *p*-values for the null hypothesis that the median of *ed*_bp_(*x*, *y*)/$K_{a}^{c}\left(x,y\right)$ is equal to 1.0 were less than 0.01 for either pair of the genotypes. The alternative hypothesis was that the median is less than 1.0. The result suggests that the constraint on the base-pairing sites is stronger than that on the non-synonymous sites. The null hypothesis that the median of *ed*_nbp_(*x*, *y*)/$K_{a}^{c}\left(x,y\right)$ is equal to 1.0 was also examined. The corrected *p*-value was less than 0.01 for either pair of the Genotypes. The alternative hypothesis for the ratios between Genotypes I and II was that the median of the ratios is less than 1.0. That is, the constraint on the non-base-pairing sites is stronger than that on the non-synonymous sites. In contrast, the alternative hypothesis for ratios between other pairs of the Genotypes was that the median is greater than 1.0. The setting of the latter alternative hypothesis was the same as that for the ratios within Genotype I or II. The results suggest that the constraint on the non-base-pairing sites is weaker than that on the non-synonymous sites.

## 4. Discussion

We examined the constraint of base pairing on the HDV genome through the statistical analyses of the distance ratios. The distance ratios, *ed*_bp_(*x*, *y*)/*ed*_nbp_(*x*, *y*), obtained from the sequence comparison within a genotype and those between a pair of genotypes were examined by the one-sample Wilcoxon test. The results suggest that the constraint on the base-pairing sites is stronger than that on the non-base-pairing sites in the non-coding region. That is, the base pairing is considered to function as a constraint on the HDV genome.

The analysis of the distance ratios of *ed*(*x*, *y*)/$K_{s}^{c}\left(x,y\right)$ suggests that the median of *ed*(*x*, *y*)/$K_{s}^{c}\left(x,y\right)$ < 1.0. Both the non-coding region and synonymous sites are considered to be free from the constraint at the protein level. Therefore, the constraint at the protein level is not considered as a cause that leads to the median of *ed*(*x*, *y*)/$K_{s}^{c}\left(x,y\right)$ < 1.0. Furthermore, to examine the cause of the difference in constraints, two distance ratios, *ed*_bp_(*x*, *y*)/$K_{s}^{c}\left(x,y\right)$ and *ed*_nbp_(*x*, *y*)/$K_{s}^{c}\left(x,y\right)$, were examined. The results suggest that not only the constraint on the base-pairing sites but also that on the non-base-pairing sites is stronger than that on the synonymous sites. The cause of constraint on the non-base-pairing sites in the non-coding region is unknown. It is suggested that the disrupted region of secondary structure, such as mismatches, bulges, and loops, may be involved in the arrangement of adenosine deaminase, the RNA editing enzyme, toward the editing site [18]. Therefore, the non-base-pairing sites may have some functional roles, which would cause the constraint on the nucleotide substitution in the non-base-pairing set. The medians of three distance ratios, *e**d*_bp_(*x*, *y*)/$K_{s}^{c}\left(x,y\right)$, *ed*_nbp_(*x*, *y*)/$K_{s}^{c}\left(x,y\right)$, and $K_{a}^{c}\left(x,y\right)$/$K_{s}^{c}\left(x,y\right),$ were compared. The order relation of the medians of the three distance ratios was identical among the comparisons within Genotype I, within Genotype II, between Genotypes I and III, and between Genotypes II and III.
$$\frac{ed_{\mathrm{bp}}\left(x,\;y\right)}{K_{s}^{c}\left(x,y\right)}<\frac{K_{a}^{c}\left(x,y\right)}{K_{s}^{c}\left(x,y\right)}<\frac{ed_{\mathrm{nbp}}\left(x,\;y\right)}{K_{s}^{c}\left(x,y\right)}<1.0$$

The order relation was also supported by the paired-sample Wilcoxon test for every possible pair of the three distance ratios (data not shown). In addition, the result of the one-sample Wilcoxon tests of *ed*_bp_(*x*, *y*)/$K_{a}^{c}\left(x,y\right)$ and *ed*_nbp_(*x*, *y*)/$K_{a}^{c}\left(x,y\right)$ and the medians were consistent with the relation. In contrast, the order relation of distance ratios obtained from the comparison between Genotypes I and II was different from those described above.
$$\frac{ed_{\mathrm{bp}}\left(x,\;y\right)}{K_{s}^{c}\left(x,y\right)}<\frac{ed_{\mathrm{nbp}}\left(x,\;y\right)}{K_{s}^{c}\left(x,y\right)}<\frac{K_{a}^{c}\left(x,y\right)}{K_{s}^{c}\left(x,y\right)}<1.0$$

The paired-sample Wilcoxon test for every possible pair of the distance ratios supported the order relation (data not shown). The results of the one-sample Wilcoxon test of *ed*_bp_(*x*, *y*)/$K_{a}^{c}\left(x,y\right)$ *ed*_nbp_(*x*, *y*)/$K_{a}^{c}\left(x,y\right)$ and the medians were consistent with the order relation.

The order relation of the distance ratios suggests that the constraint by base pairing is stronger than that by amino acid substitution. That is, the effect of negative selection of the mutations that disrupt the base pairings is greater than that of the mutations that replace the encoded amino acids. The constraint on the non-base-pairing sites is weaker than that on the base-pairing sites. In many cases, the order relation suggests that the constraint on the non-base-pairing sites is weaker than that on the non-synonymous sites, although the order relation was inverted for the ratios calculated from the comparison between Genotypes I and II.

In this study, we developed a method to detect the difference in constraint between a pair of the different regions of the HDV genome, in order to examine the effect of the base pairing on the nucleotide substitutions. As described above, the method was effective to detect the difference in constraint by base pairing between a pair of aligned regions. We conclude this manuscript with the possibility of improvement of our approach. One of the points for the improvement is the independence of the samples of distance ratios. The distance ratios used in this study were not always independent, because of the evolutionary relationship among the sequence data. The problem could be solved by calculating the ratio at each branch of the molecular phylogenetic tree or selecting the independent sequence pairs in the phylogenetic tree. A large number of genome sequences of HDV have been accumulated. Associated with the accumulation of genome sequences and other information, a database specialized for HDV has been developed [45]. However, the number is still not enough to solve the problem, because many sequences are closely related and often include unidentified bases. The other point for improvement concerns recombination. We neglected recombination in this analysis. However, it is suggested that recombination is one of the important factors to introduce diversity into HDV genomes [28]. The detection of the difference in constraint by base pairing could be improved by removing the sequences with recombination from the alignment.

## Figures and Tables

**Figure 1 viruses-13-02350-f001:**
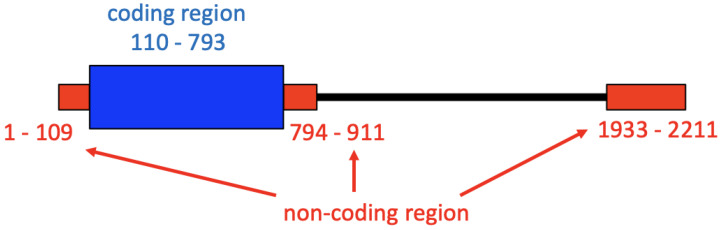
Schematic diagram of the aligned HDV genome. The three red rectangles indicate the non-coding regions. The numbers colored red indicate the positions of the non-coding regions in the alignment shown in Appendix A. The blue rectangle indicates the negative strand coding region of the L-HDAg gene. The numbers colored blue indicate the positions of the coding region in the alignment. The black line indicates the region that includes the complementary sites of the coding region.

**Figure 2 viruses-13-02350-f002:**
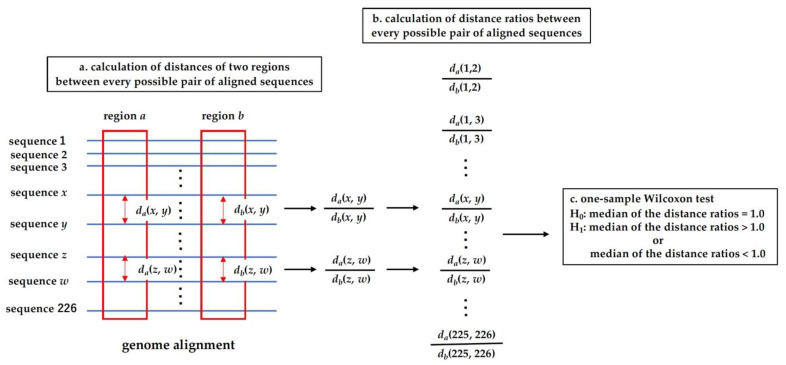
Procedure of the distance ratio analysis. (**a**) Calculation of distances of two regions between every possible pair of aligned sequences. The symbols, *d_a_*(*x*, *y*) and *d_b_(**x*, *y*), indicate the evolutionary distance between sequences *x* and *y* in region *a* and that in region *b*. The distances *d_a_*(*x*, *y*) and *d_b_*(*x*, *y*) correspond to one of the five distances *ed*_bp_(*x*, *y*), *ed*_nbp_(*x*, *y*), *ed*(*x*, *y*), $K_{s}^{c}\left(x,y\right),$ and $K_{a}^{c}\left(x,y\right)$. For simplicity of the explanation, the two regions are drawn as separate regions in an alignment. However, the regions can be overlapped. For example, when distance ratios of $K_{a}^{c}\left(x,y\right)$ and $K_{s}^{c}\left(x,y\right)$ are examined as *d_a_*(*x*, *y*) and *d_b_(**x*, *y*), the regions *a* and *b* are the non-synonymous and synonymous sites of the same coding region. (**b**) Calculation of distance ratios between every possible pair of aligned sequences. If *d_b_(**x*, *y*) equals zero, the corresponding distance ratio is not included in the set of the distance ratios. (**c**) One-sample Wilcoxon test. There are two alternative hypotheses, the median of the ratios > 1.0 and the median of the ratios < 1.0, which correspond to the right-sided and left-sided test. One of the two alternative hypotheses was adopted based on the median of the calculated distance ratios. That is, when the median was larger than 1.0 (less than 1.0), the right-sided (left-sided) one-sample Wilcoxon test was applied to the data of distance ratios.

**Figure 3 viruses-13-02350-f003:**
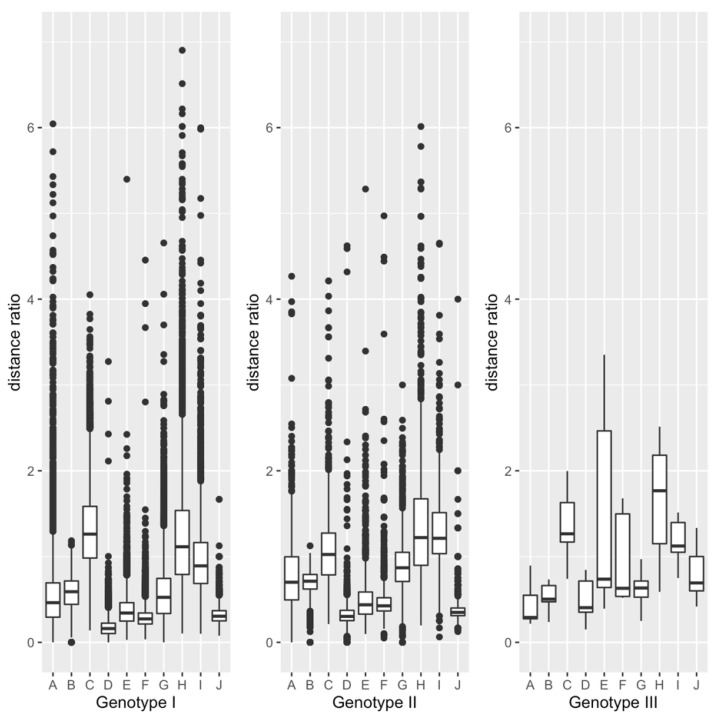
Box plots of ten distance ratios for three genotypes. The ordinate indicates the values of the distance ratios. The symbols on the abscissa, A–J, correspond to the ratios *ed*_bp_(*x*,*y*)/*ed*_nbp_(*x*, *y*), *ed*_bp_(*x*,*y*)/*ed*(*x*, *y*), *ed*_nbp_(*x*,*y*)/*ed*(*x*, *y*), *ed*_bp_(*x*,*y*)/$K_{s}^{c}\left(x,y\right)$, *ed*_nbp_(*x*,*y*)/$K_{s}^{c}\left(x,y\right)$, *ed*(*x*,*y*)/$K_{s}^{c}\left(x,y\right)$, *ed*_bp_(*x*,*y*)/$K_{a}^{c}\left(x,y\right)$, *ed*_nbp_(*x*,*y*)/$K_{a}^{c}\left(x,y\right)$, *ed*(*x*,*y*)/$K_{a}^{c}\left(x,y\right)$, and $K_{a}^{c}\left(x,y\right)$ /$K_{s}^{c}\left(x,y\right)$. The top and the bottom of a box indicates the upper and lower quantile of the distance ratios. The horizontal bar in the box indicates the median of the distance ratios. The top and the bottom of the line running through the box indicates the maximum and minimum of the ratios, except for the outliers. The ratios that are present outside a closed interval, [lower quantile−1.5×(upper quantile−lower quantile), upper quantile+1.5×(upper quantile−lower quantile)] are regarded as the outliers. The circles above or below the line indicate the outliers.

**Figure 4 viruses-13-02350-f004:**
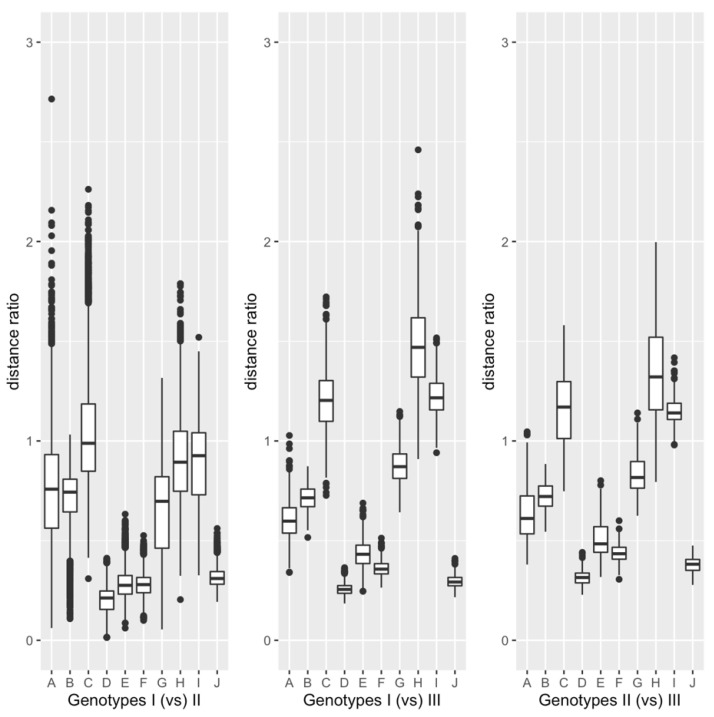
Box plots of the ten distance ratios between genotypes. The ordinate indicates the values of the distance ratios. The symbols on the abscissa, A–J, correspond to *ed*_bp_(*x*,*y*)/*ed*_nbp_(*x*, *y*), *ed*_bp_(*x*,*y*)/*ed*(*x*, *y*), *ed*_nbp_(*x*,*y*)/*ed*(*x*, *y*), *ed*_bp_(*x*,*y*)/$K_{s}^{c}\left(x,y\right)$, *ed*_nbp_(*x*,*y*)/$K_{a}^{c}\left(x,y\right)$, *ed*(*x*,*y*)/$K_{a}^{c}\left(x,y\right)$, *ed*_bp_(*x*,*y*)/$K_{a}^{c}\left(x,y\right)$, *ed*_nbp_(*x*,*y*)/$K_{s}^{c}\left(x,y\right)$, *ed*(*x*,*y*)/$K_{s}^{c}\left(x,y\right)$, and $K_{a}^{c}\left(x,y\right)$/$K_{s}^{c}\left(x,y\right)$. See legend of Figure 3 for the explanation of box plot.

**Table 1 viruses-13-02350-t001:** List of GenBank IDs for HDV used in this study.

**Genotype I (170 Sequences)**
HQ005371, MN984413, MN984411, MN984408, MN984443, MN984429, KJ744224, MN984407, HQ005367, MN984422, MN984459, MN984449, MN984415, MN984453, MN984452, MN984455, MN984451, MN984450, MN984454, MN984427, MN984437, MN984424, MN984428, MN984457, MN984448, MN984425, MN984447, MN984421, MN984430, MN984423, MN984461, MN984414, MN984432, MN984410, MN984431, MT583796, MN984412, MN984460, MN984416, KJ744217, KJ744216, KJ744215, KJ744214, MN984435, MN984436, MN984409, MN984445, MN984419, MN984441, AB118848, MN984440, MN984438, MN984442, MN984417, MN984439, MN984420, MN984466, MN984458, MH457143, X85253, MT583812, KJ744223, KJ744220, KJ744222, KJ744221, KT722840, MN984456, KF660602, AB118849, MN984426, AY648957, AY648956, KF660600, MN984444, MN984434, AY648959, AF425644, AY648958, AF104263, MH791030, MH791028, KY463681, U81989, MG926381, MK890226, MK890225, MK890231, HQ005372, MG926380, HQ005366, MN984469, MH791029, MK890228, MK890227, MK890232, MK890235, KJ744237, MK890234, HQ005370, HQ005364, MK890230, HQ005368, HQ005365, U81988, AY633627, KJ744255, MN984418, KJ744238, MH791027, MT583805, MT583804, MH457145, KJ744243, MN984463, MN984433, KJ744257, KJ744244, EF514905, EF514907, EF514904, EF514903, EF514906, KJ744256, MN984465, KJ744228, KJ744227, KJ744234, KJ744254, KJ744248, KJ744232, KJ744253, KJ744247, MN984462, KJ744245, MN984464, KJ744218, KJ744249, KJ744231, MN984446, KJ744226, KJ744225, NC_001653, D01075, KJ744230, KJ744229, KJ744235, KJ744233, KJ744250, MK124579, M21012, HM046802, AJ307077, AJ000558, HQ005369, KJ744242, KJ744240, KJ744241, MK890224, MK890229, MG711778, MK890233, KM110794, KM110792, KM110797, KM110799, MG711717, KM110795, KM110791, KM110798, KM110790
**Genotype II (51 sequences)**
KF660599, KF660598, MG557658, AB118846, AY261457, AF425645, AF104264, MK234591, MK234593, MK234592, MK234594, MG557659, MN984468, KM110805, MN984470, MG711777, AB118844, AB118842, AF209859, MT050453, AB118843, AY648954, AY648953, AY648952, AB118847, AY648955, AB118841, MN401236, AB118845, AB118826, AB118835, AB118834, AB118819, AB118828, AB118832, AB118840, AB118830, AB118827, AB118820, AB118825, AB118821, AB118839, AB118822, AB118818, AB118833, AB118838, AB118823, AB118836, AB118824, AB118837, AB118831
**Genotype III (5 sequences)**
AB037947, KC590319, HF679406, HF679405, HF679404

**Table 2 viruses-13-02350-t002:** Average ratio of bases that bind to other bases.

Threshold	Average Ratio
0.4	0.753
0.5	0.722
0.6	0.672
0.7	0.623

**Table 3 viruses-13-02350-t003:** Medians and corrected *p*-values of ten distance ratios for three genotypes. The size of the samples of each genotype is indicated by *n*. The corrected *p*-values were calculated by the left-sided test (the alternative hypothesis is that the median is less than 1.0) or the right-sided test (the alternative hypothesis is that the median is greater than 1.0.). The asterisk indicates the *p*-value calculated by the right-sided test. The floating point number representation is used to express the *p*-values.

		*ed*_bp_(*x*,*y*)/*ed*_nbp_(*x*, *y*)	*ed*_bp_(*x*,*y*)/*ed*(*x*, *y*)	*ed*_nbp_(*x*,*y*)/*ed*(*x*, *y*)	*ed*_bp_(*x*,*y*)/$K_{s}^{c}\left(x,y\right)$	*ed*_nbp_(*x*,*y*)/$K_{s}^{c}\left(x,y\right)$	*ed*(*x*,*y*)/$K_{s}^{c}\left(x,y\right)$	*ed*_bp_(*x*,*y*)/$K_{a}^{c}\left(x,y\right)$	*ed*_nbp_(*x*,*y*)/$K_{a}^{c}\left(x,y\right)$	*ed*(*x*,*y*)/$K_{a}^{c}\left(x,y\right)$	$K_{a}^{c}\left(x,y\right)$ $K_{s}^{c}\left(x,y\right)$
**I**	**median**	0.46	0.59	1.3	0.16	0.34	0.27	0.52	1.1	0.89	0.30
*n* = 14,276	***p*-value**	<2.2 × 10^−15^	<2.2 × 10^−15^	<2.2 × 10^−15^ *	<2.2 × 10^−15^	<2.2 × 10^−15^	<2.2 × 10^−15^	<2.2 × 10^−15^	<2.2 × 10^−15^ *	<2.2 × 10^−15^	<2.2 × 10^−15^
**II**	**median**	0.70	0.71	1.0	0.30	0.44	0.43	0.87	1.2	1.2	0.35
*n* = 1263	***p*-value**	<2.2 × 10^−15^	<2.2 × 10^−15^	8.6 × 10^−4^ *	<2.2 × 10^−15^	<2.2 × 10^−15^	<2.2 × 10^−15^	<2.2 × 10^−15^	<2.2 × 10^−15^ *	<2.2 × 10^−15^ *	<2.2 × 10^−15^
**III**	**median**	0.29	0.50	1.3	0.40	0.74	0.63	0.63	1.8	1.1	0.69
*n* = 9	***p*-value**	1.9 × 10^−2^	1.9 × 10^−2^	1.4 × 10^−1^ *	2.0 × 10^−2^	1.0 *	1.0 *	2.0 × 10^−2^	2.0 × 10^−1^ *	6.4 × 10^−1^ *	1.0

**Table 4 viruses-13-02350-t004:** Medians and corrected *p*-values of ten distance ratios between genotypes. The size of the samples of each genotype is indicated by *n*. See legend of Table 3 for the asterisk and the representation of the *p*-value.

		*ed*_bp_(*x*,*y*)/*ed*_nbp_(*x*, *y*)	*ed*_bp_(*x*,*y*)/*ed*(*x*, *y*)	*ed*_nbp_(*x*,*y*)/*ed*(*x*, *y*)	*ed*_bp_(*x*,*y*)/$K_{s}^{c}\left(x,y\right)$	*ed*_nbp_(*x*,*y*)/$K_{s}^{c}\left(x,y\right)$	*ed*(*x*,*y*)/$K_{s}^{c}\left(x,y\right)$	*ed*_bp_(*x*,*y*)/$K_{a}^{c}\left(x,y\right)$	*ed*_nbp_(*x*,*y*)/$K_{a}^{c}\left(x,y\right)$	*ed*(*x*,*y*)/$K_{a}^{c}\left(x,y\right)$	$K_{a}^{c}\left(x,y\right)/K_{s}^{c}\left(x,y\right)$
**I (vs.) II**	**median**	0.76	0.74	0.99	0.21	0.28	0.28	0.70	0.89	0.93	0.31
*n* = 8670	***p*-value**	<2.2 × 10^−15^	<2.2 × 10^−15^	<1.8 × 10^−7^ *	<2.2 × 10^−15^	<2.2 × 10^−15^	<2.2 × 10^−15^	<2.2 × 10^−15^	<2.2 × 10^−15^	<2.2 × 10^−15^	<2.2 × 10^−15^
**I (vs.) III**	**median**	0.60	0.71	1.2	0.26	0.43	0.36	0.87	1.5	1.2	0.29
*n* = 850	***p*-value**	<2.2 × 10^−15^	<2.2 × 10^−15^	<2.2 × 10^−15^ *	<2.2 × 10^−15^	<2.2 × 10^−15^	<2.2 × 10^−15^	<2.2 × 10^−15^	<2.2 × 10^−15^ *	<2.2 × 10^−15^ *	<2.2 × 10^−15^
**II (vs.) III**	**median**	0.61	0.72	1.2	0.32	0.48	0.43	0.82	1.3	1.1	0.38
*n* = 255	***p*-value**	<2.2 × 10^−15^	<2.2 × 10^−15^	<2.2 × 10^−15^ *	<2.2 × 10^−15^	<2.2 × 10^−15^	<2.2 × 10^−15^	<2.2 × 10^−15^	<2.2 × 10^−15^ *	<2.2 × 10^−15^ *	<2.2 × 10^−15^

## Data Availability

The data is contained within the article or Appendix A.

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
