# Peer review of "Constraint of Base Pairing on HDV Genome Evolution"

_viruses, 2021, doi:10.3390/v13122350_

Round 1
Reviewer 1 Report
1. The novel part of this paper is the method. However, it is hard to justify why this method serves the purpose. It seems that the author tried to develop a complicated method to probe a simple question. Can the author identify a region which does not often form a stem structure (a mapping using enzymatic digestions should generate such regions) and then probe the same question? In all, the author needs to justify their method? since it is very hard to predict RNA structures, any unproven prediction method needs a justification.
2. It is extremely hard to understand the method part. The author needs to do a better writing, maybe using some figures. No need to waste space for Figure 1.
Author Response
Responses to Reviewer 1
First of all, we express our thankfulness for the comments. All the comments were reflected in the revised manuscript.
Comment 1: The novel part of this paper is the method. However, it is hard to justify why this method serves the purpose. It seems that the author tried to develop a complicated method to probe a simple question. Can the author identify a region which does not often form a stem structure (a mapping using enzymatic digestions should generate such regions) and then probe the same question? In all, the author needs to justify their method? since it is very hard to predict RNA structures, any unproven prediction method needs a justification.
Our response to comment 1
In the original version, we had used complicated approach as the reviewer pointed out. One of the reasons is that we had wanted to improve the accuracy of secondary structure prediction by using the multiple alignment. However, it had made the procedure difficult to understand. The key point of the procedure is the one-sample Wilcoxon test. So, we re-organized the procedure based on the pairwise comparison (lines 178-189). The correction of multiple comparison was replaced from the Benjamini-Hochberg method to Bonferroni method for simplicity (lines 273-276). The distance ratio analysis is divided into three steps, and explained the details step by step (lines 202 – 267 and Figure 2). Each step is explained with the corresponding subfigure in Figure 1.
We are afraid that the reviewer misunderstood that we used an unauthorized method for the secondary structure prediction. When we had developed a version of MAFFT to incorporate secondary structure prediction information (reference 32), the McCaskill algorithm (reference 33) implemented in Vienna RNA package (reference 34) was incorporated. The correspondence with the cleavage sites, which the reviewer pointed out, is discussed in reference 33. We clearly described that the McCaskill algorithm is incorporated in MAFFT and cited the paper of McCaskill algorithm and Vienna RNA package in the revised manuscript (lines 93 – 98).
Comment 2: It is extremely hard to understand the method part. The author needs to do a better writing, maybe using some figures. No need to waste space for Figure 1.
Our response to comment 2
I omitted the Figure 1 of original version. As described in the response to comment 1, the method is explained step by step with new Figure 2.
Reviewer 2 Report
This review describes the main features of the hepatitis delta virus, a small RNA virus that needs the hepatitis B virus to propagate intercellularly.
Despite its identification goes back to the mid-seventies, little is known about its RNA genome and interaction with hepatitis B virus, from which it borrows the surface antigen. This review provides a method to measure the evolutionary rate based on base-pairing models in the non-coding region of the virus. Of interest, this method also suggests that the non-coding nucleotides without base-pairing are not so free to change as originally thought.
The manuscript is interesting, easy to read and provides a novel method to infer the genetic evolution of viruses. Unfortunately, the figures are of a lower standard. For instance, it is very difficult to distinguish the red and blue lines of Figure 2 and captions do not explain what the Figures show forcing to zap between text and figures to understand their meaning.
The caption of Figure 1 is not clear; please explain better how the non-coding and complementary regions are located in the HDV genome. Insert a schematic model of the HDV genome if necessary.
Please describe better what Figures 2 and 4 show and provide a legend in Figure 3.
Line 359, replace HDB with HDV.
Author Response
We thank the reviewer for the comments. We revised the manuscript to reflect the comments.
Comment 1 : Unfortunately, the figures are of a lower standard. For instance, it is very difficult to distinguish the red and blue lines of Figure 2 and captions do not explain what the Figures show forcing to zap between text and figures to understand their meaning.
Our response to comment 1
Figure 2 is thoroughly redrawn in the current version. The distance ratio analysis is divided into three steps, and explained the details step by step (lines 202 – 267 and Figure 2). Each step is explained with the corresponding subfigure in Figure 2. The information about the subfigures is described in the legend of Figure 2.
Comment 2 : The caption of Figure 1 is not clear; please explain better how the non-coding and complementary regions are located in the HDV genome. Insert a schematic model of the HDV genome if necessary.
Our response to comment 2
We replaced the original Figure 1 to explain the positions used in this study. We also provided an alignment data used in this study as Supplementary data 1 (FASTA format). The coding and non-coding regions are explicitly described as the positions in the alignment (Supplementary data 1) in lines 149 – 155.
Comment 3: Please describe better what Figures 2 and 4 show and provide a legend in Figure 3.
Our response to comment 3
As described in our response to Comment 1, Figure 2 is thoroughly re-drawn.
(see Our response to comment 1 for more details).
Figure 3 is omitted in the revised manuscript.
Figure 4 is re-organized as Figures 3 and 4 in the revised manuscript. The w ratio is shown in the lane J of each subfigure. Both Figures 3 and 4 have legends for the explanation.
Comment 4 : Line 359, replace HDB with HDV.
Our response to comment 4
Thank you very much for the comment. We corrected HDB with HDV (line 474).
Reviewer 3 Report
In their paper „Constraint of Base-Pairing on HDV Genome Evolution”, the authors perform an analysis on how mutation rates of the hepatitis delta virus circular genome are governed or impeded by the requirement for base pairing within the rod-like genome structure. To this end, the authors perform a scoring of the base pairing within the genome for each nucleotide, and compare the constraints on nucleotide changes in various regions of the genome. They find that there indeed are differences, and that their approach is able to detect said differences.
While the ideas are generally interesting, the paper falls short in their execution as well as their description.
As a general remark, I found the paper difficult to read and comprehend for a virology or RNA-oriented readership. Key information is simply missing: a description of where the boundaries for the identified regions were, how many nucleotides belonged to each region or category, a quantification of the differences (or better similarity) between the genomes, just to name a few. These would be important to actually judge the mutation rate and the constraints described in this manuscript.
As for illustration, I find the figures to be little informative. The coloring of Figures 1 and 2 is misleading, since identical colors depict different things, and one cannot derive usable information (i.e. where in the genome is the coding region, etc.). The same applies to Figure 3, where an actual illustration of the genome would be helpful. Here, the illustration is misleading in that it does not reflect the ~70% of basepaired nucletides, but rather indicates a higher percentage due to the limited resolution.
The values for Ks and Ka should be given, and not only omega ratios, in order to render this part of the manuscript better comprehendable. The description of the results regarding differences between the clade are not sufficiently detailed.
Overall, the results are somewhat expected and give only incremental if any knowledge increase. On the other hand, in order to establish their approach as a new method, it would be necessary to provide a much more detailed description of the approach that would need to be easier to reproduce.
In the end, I have two major criticisms on a broader scope. First of all, of all the manuscript’s citations only about 10% are from the last 15 years, which raises questions to the actuality of this research. At the same time, there have been a number of publications on HDV genome structure, which are not cited, as well as a publication from more than a year ago that contains a significantly larger and substantially more informative dataset than the one used in this study (DOI: 10.3390/v12050538), which incidentally was published in this very Journal.
In summary, due to the questions regarding actuality, the shortcomings in description and illustration, and the merely incremental gain in knowledge, I cannot recommend this paper for publication.
Minor issues:
W ratio: line 286
Line 12: constraint
Author Response
We thank the reviewer for the comments. We revised the manuscript to reflect the comments.
Comment 1: As a general remark, I found the paper difficult to read and comprehend for a virology or RNA-oriented readership. Key information is simply missing: a description of where the boundaries for the identified regions were, how many nucleotides belonged to each region or category, a quantification of the differences (or better similarity) between the genomes, just to name a few. These would be important to actually judge the mutation rate and the constraints described in this manuscript.
Our responses to comment 1
We provided the information pointed out by the reviewer as follows.
We provided an alignment data used in this study as Supplementary data 1 (FASTA format). The coding and non-coding regions are explicitly described as the positions in the alignment (Supplementary data 1) in lines 149 – 155. The regions are explained in the schematic diagram of aligned HDV genome (see Figure 2).
The size of each region and the difference are described in Supplementary data 3. The difference and the correction of multiple hits are described in lines 167-200.
Comment 2 : As for illustration, I find the figures to be little informative. The coloring of Figures 1 and 2 is misleading, since identical colors depict different things, and one cannot derive usable information (i.e. where in the genome is the coding region, etc.). The same applies to Figure 3, where an actual illustration of the genome would be helpful. Here, the illustration is misleading in that it does not reflect the ~70% of basepaired nucletides, but rather indicates a higher percentage due to the limited resolution.
Our response to Comment 2: The manuscript was thoroughly rewritten. Figures were also re-drawn.
The original Figure 1 was omitted. As described in response to comment 1, we provided an alignment data as Supplementary data 1. The coding and non-coding regions are explicitly described as the positions in the alignment in lines 149 – 155. The regions are also explained in Figure 1. The positions of the coding and non-coding regions were also shown in new Figure 1, which represents the schematic diagram of HDV genome alignmen.t
Figure 2 is re-drawn in order to respond to the comments from other reviewer. The color problem pointed out by the reviewer is solved by the omission of the original Figure 1.
The original Figure 3 is omitted to avoid the confusion pointed out by the reviewer.
Comment 3 : The values for Ks and Ka should be given, and not only omega ratios, in order to render this part of the manuscript better comprehendable. The description of the results regarding differences between the clade are not sufficiently detailed.
Our response to Comment 3: The values of Ks and Ka and other evolutionary distances are given in Supplementary data 3. In the revised manuscript, the genome data were examined at the Genotype level, instead of the clade level. The distance ratios were analyzed between Genotypes, as well as within each Genotypes. They are described in lines 281 – 413.
Comment 4: In the end, I have two major criticisms on a broader scope. First of all, of all the manuscript’s citations only about 10% are from the last 15 years, which raises questions to the actuality of this research. At the same time, there have been a number of publications on HDV genome structure, which are not cited, as well as a publication from more than a year ago that contains a significantly larger and substantially more informative dataset than the one used in this study (DOI: 10.3390/v12050538), which incidentally was published in this very Journal.
Our response to Comment 4: The aim of this manuscript is not the review of the genome data. As described in the Introduction, we’d like to point out that the neglection of the secondary structure affect the evolutionary analysis. So, we did not include all the genome studies in the references, but the paper pointed out by the reviewer were included in the references. In the revised manuscript, we increased the genome data for the analysis. As described in the manuscript, many sequences included unidentified bases and/or frameshift in the coding region. Such data were not included in the analysis. The GenBank IDs of the genome data used in this study are listed in Table 1.
Reviewer 4 Report
The manuscript entitled “Constraint of Base-Pairing on HDV Genome Evolution”, submitted to Viruses by Saki Nagata, Ryoji Kiyohara and Hiroyuki Toh. The authors developed a new method based on one sample t-test to examine the effect of the base-pairing as the constraint on nucleotide substitution. The result of their analysis suggested that the base-pairing imposes a constraint on the nucleotide substitution in the non-coding region of HDV. It is also suggested that the nucleotides in the non-coding region which are not involved in the base-pairing may be under some constraint. The intensity of the constraint on the non-coding region without base-pairing is weaker than that on the non-coding region with base-pairing, but stronger than that on the synonymous site.
Specific comment:
(1) Page 3, lines 119-132.
The authors developed a new method to examine the effect of base-pairing as the constraint on nucleotide substitution. Thus, the reader at least needs to understand “the base-pairing probability”. The authors should provide more information on “the base-pairing probability”.
(2) Page 3, lines 123 and 133-151.
The authors described “the nucleotide i” and “HDV-i”. However, the former “i” is different from the latter “i “. Their descriptions will confuse the readers. The authors should change either.
(3) Page 6, Figure 3.
Figure 3 will confuse the readers because the axis labels and color bar are not shown in this heatmap figure. Why is the background light blue? The authors should provide more information for the readers to understand the figure.
(4) Page 7, Table 2.
For example, nc_diff2/Ks, nc_diff3/Ks, and nc_diff3/nc_diff2 in the column of 60% of HDV2 are 0.52, 0.93, and 3.11, respectively. The nc_diff3/nc_diff2 is not equal to (nc_diff3/Ks)/(nc_diff2/Ks). It will be helpful for the reader if the authors provide a brief comment this issue.
Minor comment:
(1) Page 6, line 211.
Although R is the language and environment for statistical computing, the readers do not always know the information. It will be helpful for the reader if the authors provide more information on R.
Author Response
We thank the reviewer for the comments. We revised the manuscript to reflect the comments.
Comment 1: Page 3, lines 119-132.
The authors developed a new method to examine the effect of base-pairing as the constraint on nucleotide substitution. Thus, the reader at least needs to understand “the base-pairing probability”. The authors should provide more information on “the base-pairing probability”.
Our response to Comment 1: In the original version, we omitted many things about the secondary structure prediction. MAFFT uses the McCaskill algorithm implemented in the Vienna RNA package for the secondary structure prediction. In the original version, we just mentioned MAFFT, but missed to describe the algorithm. In addition to MAFFT, we described the algorithm in the revised manuscript. According to the description in the paper of McCaskill algorithm, the calculation method of the base-pairing probability is briefly described in the revised manuscript (lines 97-100).
Comment 2: Page 3, lines 123 and 133-151.
The authors described “the nucleotide i” and “HDV-i”. However, the former “i” is different from the latter “i “. Their descriptions will confuse the readers. The authors should change either.
Our response to Comment 2 : In the revised manuscript, the genome data were examined at the Genotype level, instead of the clade level. Accompanied with the change, the expression HDV-i is removed.
Comment 3: Page 6, Figure 3.
Figure 3 will confuse the readers because the axis labels and color bar are not shown in this heatmap figure. Why is the background light blue? The authors should provide more information for the readers to understand the figure.
Our response to Comment 3 : Thank you very much for the comment. Figure 3 is not informative for the distance ratio analysis. So, we removed Figure 3 from the revised manuscript.
Comment 4 : (4) Page 7, Table 2.
For example, nc_diff2/Ks, nc_diff3/Ks, and nc_diff3/nc_diff2 in the column of 60% of HDV2 are 0.52, 0.93, and 3.11, respectively. The nc_diff3/nc_diff2 is not equal to (nc_diff3/Ks)/(nc_diff2/Ks). It will be helpful for the reader if the authors provide a brief comment this issue.
Our response to Comment 4 : Thank you very much for pointing out the problem. The numbers above the q-values in the original Table 2 indicated the means of the distance ratios. As described above, we thoroughly re-analyzed the data and the statistical test was changed from one-sample t-test to one-sample Wilcoxon test, because the distance ratios did not seem to follow the normal distribution. So, instead of means, the medians of distance ratios are shown in Tables 3 and 4. The meanings of the numbers shown in the tables are described in the legends.
Comment 5 : Page 6, line 211.
Although R is the language and environment for statistical computing, the readers do not always know the information. It will be helpful for the reader if the authors provide more information on R.
Our response to Comment 5 : Brief explanation is introduced in lines 272-273. According to the description in CRAN, the citation of R is added in the references.
Round 2
Reviewer 1 Report
It is extremely hard to predict RNA secondary structures using any available methods. Any conclusions made using such predictions needs to be verified alternatively or experimentally. It is very easy to do that nowadays based on partial digestion and high-throughput sequencing.
Reviewer 3 Report
In their paper „Constraint of Base-Pairing on HDV Genome Evolution”, Nagata et al describe a method to identify evolutionary constraint on HDV genome regions that are involved in basepairing.
As pointed out by Reviewer #1, the paper essentially describes a circular conclusion: there is evolutionary constraint on basepairing and sequence, and these two findings are dependent on each other. Their described method essentially provides proof of this finding, but offers no new information beyond that, such as which regions of the genome are structured, which regions are conserved, what would be functional implications of such findings.
My previous comments on Figures have been taken into account in the revisions, but the major criticisms have remained largely unaddressed. In their introduction, the authors state that “The result of our analysis suggests that the base-pairing imposes constraint on the nucleotide substitution in the non-coding region of HDV”. This is completely expected, and not novel.
The authors provide no attempt to link their findings to functional relevance (i.e., are there key functional elements of the genome that are under unexpectedly high or low constraint?). The revisions, although extensive, have not corrected for the major flaws.
Overall, I find that paper to be lacking findings of sufficient novelty, and cannot recommend publication.